# The Transverse Mechanical Axis of the Pelvis for Post-Operative Evaluation of Total Hip Arthroplasty

**DOI:** 10.3390/biomedicines11051397

**Published:** 2023-05-08

**Authors:** Cheng-Jui Tsai, Zong-Yan Yang, Tai-Yin Wu, Ya-Ting Tsai, Juyn-Jhe Wang, Chen-Kun Liaw

**Affiliations:** 1School of Medicine, College of Medicine, Taipei Medical University, Taipei City 11031, Taiwan; 2Department of Family Medicine, Zhongxing Branch, Taipei City Hospital, Taipei City 10341, Taiwan; 3Institute of Epidemiology and Preventive Medicine, National Taiwan University, Taipei City 10055, Taiwan; 4General Education Center, University of Taipei, Taipei City 10023, Taiwan; 5School of Medicine, China Medical University, Taichung City 404333, Taiwan; 6Department of Orthopedics, School of Medicine, College of Medicine, Taipei Medical University, Taipei City 11031, Taiwan; 7Department of Orthopedics, Shuang Ho Hospital, Taipei Medical University, No. 291, Zhongzheng Rd., Zhonghe Dist, New Taipei City 23561, Taiwan; 8Research Center of Biomedical Device, College of Biomedical Engineering, Graduate Institute of Biomedical Optomechatronics, Taipei Medical University, Taipei City 11031, Taiwan; 9TMU Biodesign Center, Taipei Medical University, Taipei 11031, Taiwan

**Keywords:** total hip arthroplasty, acetabular components, pelvic axis, postoperative evaluation, pelvis anteroposterior radiographs, limb length discrepancy (LLD), hip rotation center, acetabular teardrop, obturator foramen, iliac crest, ischial tuberosity

## Abstract

Currently, orthopedic surgeons mainly use the inter-teardrop line (IT-line) as the transverse mechanical axis of the pelvis (TAP) for postoperative evaluation of total hip arthroplasty (THA). However, the teardrop is often unclear in the pelvis anteroposterior (AP) radiographs, which makes postoperative evaluation of THA difficult. In this study, we attempted to identify other clear and accurate axes for postoperative evaluation of THA. We calculated the mean and standard deviation of these angles and tested the significance of these angles using *t*-tests. The inter-teardrops line (IT line) and the upper rim of the obturator foramen (UOF) had smaller angles with the IFH line. The bi-ischial line (BI line) was relatively inaccurate in measurements. We recommend using the IT line as the TAP when the lower boundary of the teardrops is clear and the shapes of the teardrops on both sides of the pelvis are symmetrical. When there is no deformation of the obturator foramen on pelvic AP radiographs, the UOF is also a good choice for the TAP. We do not recommend the BI line as the TAP.

## 1. Introduction

Total hip arthroplasty (THA) is regarded as one of the most successful and cost-effective interventions in modern orthopedics [1,2,3]. The number of THA procedures performed worldwide has exponentially increased over the past decade [3]. For patients with advanced arthritis, it relieves pain and improves the patient’s mobility and quality of life [2,4]. Although the procedure is effective and frequently performed, if the acetabular cup is not placed in the proper position, certain postoperative complications such as impingement [5], dislocation [5], and decreased durability of the acetabular cup [6], poor functional outcomes [7], and general dissatisfaction of the patients [8] might occur. Therefore, a comprehensive postoperative evaluation of THA is critical to manage the risks and complications of the patients. Rehabilitators and physical therapists may also develop suitable rehabilitation plans based on the results of the postoperative evaluation.

The transverse mechanical axis of the pelvis (TAP) can help physicians evaluate the pelvic inclination angle [6,9], locate the hip rotation center [6,10], and assess the severity of the limb length discrepancy (LLD) after THA [11,12]. These parameters can affect the prognosis of the patients, the durability of the acetabular cups, and the occurrence of related complications [5,6,8,13]. Improper acetabular inclination may cause impingement and subsequent dislocation [5]. High correlation between the acetabular inclination and the wear direction was also noted [13,14,15]. Under normal circumstances, the hip rotation center is the center of the femoral head [16]. However, after undergoing THA, the hip rotation center will undergo changes [16,17,18]. The hip rotation center is related to the hip joint reactive forces and the wear of the acetabular cup [6,19]. A medialized, inferior, and anterior hip rotation center will reduce the reactive joint forces and therefore improve wear mechanics [6]. A normal or slight inferomedial hip rotation center can significantly increase the hip abductor strength [6]. LLD is associated with back pain and sciatica, neuritis, gait disorders, general dissatisfaction, dislocation, and early loosening of components [8]. Nearly 50% of patients have LLD ≥ 1 cm after THA, and of which 15–20% of patients require corrective shoes to equalize the limb lengths [8]. Not all limb length discrepancy will cause obvious discomfort to the patient, but LLD > 2 cm tends to be more problematic [20,21,22,23]. An accurate TAP can help physicians evaluate the above problems, assess the severity of the patients, and then provide appropriate follow-up treatments.

Orthopedic surgeons mainly use the inter-teardrops line (IT line) to represent the TAP [24,25,26]. The acetabular teardrop is a radiographic feature located in the anteroinferior portion of the acetabular fossa at the acetabular notch on the pelvis anteroposterior (AP) radiographs [27,28]. The teardrop was selected as the anatomic landmark of the pelvis due to its high consistency and reliability in measurement [9,24]. The difference in vertical distance from the teardrops to lesser trochanters on both sides of the lower extremities is used as the radiological method to measure LLD [11,12,29]. The hip rotation center can be located using the teardrop as a reference point [18,30]. The pelvic abduction (inclination) angle can be measured through the included angle between the IT line and the line formed by the connection of the lower edge of the teardrop and the femoral head center [13]. Using the teardrop as an anatomic landmark, orthopedic surgeons can evaluate the relative position between the pelvis and the acetabular cup after THA [26,31]. Figure 1 illustrates the current method of using TAP to measure LLD.

Although the IT line shows good accuracy as measuring the TAP, the teardrop or its lower edge is often not visible in pelvis AP radiographs. Clinically, which anatomical landmark should be used as an alternative of the teardrop remains inconclusive. The lack of a clear TAP creates difficulties for orthopedic surgeons to assess the postoperative condition of THA patients. Orthopedic surgeons sometimes use other TAPs for measurements, but there is no comprehensive research explaining the accuracy of these TAPs. We conducted this study to identify other clear and accurate axes to represent the TAP for postoperative evaluation and further development of rehabilitation plans for THA.

## 2. Materials and Methods

We reviewed 120 pelvis AP radiographs from 1 November 2022 to 31 December 2022 at Taipei Medical University Shuang-Ho Hospital. In order to determine which TAP could best measure the position of the femoral head under normal circumstances, we excluded patients with obvious abnormalities or with acetabular implants, and this is because severe arthritis or artificial placement of acetabular implants may both cause changes to the hip rotation center. Our exclusion criteria included patients under the age of 20 (n = 9), images without clear teardrops (n = 94), images without completely presented iliac crests (n = 22), patients with acetabular implants (n = 36), patients with significant deformation of the anatomy of the pelvis or femoral head (pelvic fractures, bone tumor, severe femoral fractures or osteoarthritis that affect the shape or the position of the femoral head) (n = 15), and images where the obturator foramen was without clear boundaries. (n = 2). Figure 2 is the flow diagram of our recruitment process. This study was approved by the Taipei Medical University—Joint Institutional Review Board (approval number N202209057).

Due to the need to compare the relative positions between each TAP and the femoral head, and since the femoral head center is the hip rotation center under normal conditions, we set the reference axis as the inter-femoral-head-centers line (IFH line). The following five axes, the supracristal line (SC line), bi-ischial line (BI line), inter-teardrops line (IT line), and the upper and lower rim of the obturator foramen (UOF and LOF), were selected as candidates representing the TAP. The reference axis and the other five axes were drawn by connecting the corresponding anatomic landmarks on both sides of the pelvis. The IFH line was defined as the line connecting the femoral head centers. The center of the femoral head was located by drawing a circle that fits the femoral head. The SC line was defined as the line connecting the highest points of both the iliac crests. The BI line was defined as the line connecting the ischial tuberosities. The IT line was defined as the line connecting the lower edges of the teardrops. The UOF was defined as the line connecting the highest points of the obturator foramens. The LOF was defined as the line connecting the lowest points of the obturator foramens. It is worth mentioning that the exact upper border of the obturator foramen may be obscured by the bone in front. The measurer should carefully identify the true upper border of the obturator foramen based on the brightness variation of its border on the image in order to trace the UOF accurately. (Figure 3).

Our self-developed program “PelvicAxis” was used to measure the included angles between the reference axis (the IFH line), and the pelvic axes (Figure 4). Since we had studied five pelvic axes in total, we will get five included angles for each measurement, namely the included angles between the reference axis and the SC line (∠RSC), between the reference axis and the BI line (∠RBI), between the reference axis and the IT line (∠RIT), between the reference axis and the UOF (∠RUO), and between the reference axis and the LOF (∠RLO). Based on the reference axis, clockwise rotation was defined as positive and counterclockwise rotation was defined as negative. The measurements in this study were performed by two observers. Observer A measured all 120 radiographs twice, while observer B measured them once.

Because our research primarily aimed to measure the deviations of the angles between TAPs and the reference axis, we took the absolute value of all measured values before calculating the mean and standard deviation of the angles. One-sample *t*-test was used to compare whether the included angle between an axis and the reference axis was statistically significantly. If the TAP can accurately measure the position of the femoral head under normal conditions, then the included angle should not be significantly different from 0. To compare the measurement performance of the commonly used IT line with the other four TAPs, we conducted two sample *t*-tests to compare the mean of ∠RIT with ∠RSC, ∠RBI, ∠RUO, and ∠RLO, respectively. We also divided the sample into subgroups of different genders and age ranges for analysis. For outlier analysis, we visualize the overall dispersion of the data using a boxplot and define outliers based on the distribution shown in the plot. As each subject’s data point contains five angles and the degree of dispersion among the five angles varies, we select the two data points with the highest deviation for each TAP’s included angle as our final outliers, which means we have a total of 10 outliers. We removed the outliers, conducted another analysis on the overall data, and discussed the possible reasons for the occurrence of outliers in the discussion section. Reliability analysis was used to analyze interobserver and intraobserver reliability and the outcome was presented as the intraclass correlation coefficient (ICC). As we wanted to reflect the most accurate measurement situation, we used the raw measurement data without taking the absolute value to calculate the ICC. Data were analyzed using IBM Corp. Released 2010. IBM SPSS Statistics for Windows, Version 19.0. Armonk, NY: IBM Corp.

## 3. Results

### 3.1. Basic Information before Exclusion

Overall, 94 (34.2%) radiographs had the problem of unclear teardrops, which was our main reason for exclusion. For the 36 excluded radiographs with acetabular implant, there were 13 (36%) radiographs without clear teardrops. For the 239 (=275 − 36) radiographs without acetabular implants, there were 81 (=94 − 13) (33.9%) radiographs without clear teardrops. Among all 275 radiographs, only 2 (1%) radiographs had the problem of unclear obturator foramen boundaries and such a situation did not appear on radiographs with acetabular implants.

### 3.2. Basic Information of the Subjects We Included

Our final analysis comprised 58 males and 62 females; their ages ranged from 20 to 96 years old. The average age of our subjects was 56.1 years old, and the standard deviation of our subjects’ age was 21.2 years old (Table 1).

### 3.3. Reliability Analysis

Table 2 is the intra- and interobserver correlation coefficient (ICC) in the measurements of each angle. Note that the data used to calculate ICC was the raw data without taking the absolute value. Overall, each angle had a reliability above moderate. However, the reliability of ∠RIT and∠RUO were lower than the other three angles.

### 3.4. The Measurement Results for All Subjects

Table 3 is the result of the included angles between the five pelvic axes and the IFH line measured by the two observers of all subjects. Note that the data used to calculate the mean and standard deviation below had been absolute-valued before calculation. The data are presented as mean ± standard deviation. The means and standard deviations of ∠RIT and ∠RUO were smaller than the other three angles. The means of ∠RBI were the largest among the five angles. All the angles measured by the two observers were significantly different from 0. From Table 4, it can be seen that there was no significant difference between the means of ∠RUO and ∠RIT, while there were significant differences between ∠RIT and the other three angles.

### 3.5. The Measurement Results for Female Subjects

Table 5 and Table 6 are the results for the female subjects. The means and standard deviations of ∠RIT and ∠RUO were smaller than the other three angles. The means of ∠RBI were the largest among the five angles. All the angles measured by the two observers were significantly different from 0. The means of ∠RUO and ∠RIT showed no significant differences in all three measurements, ∠RSC and ∠RIT showed no significant differences in two measurements, while ∠RBI and ∠RLO both showed significant differences compared with ∠RIT.

### 3.6. The Measurement Result for Male Subjects

Table 7 and Table 8 are the results for the male subjects. The means and standard deviations of ∠RIT and ∠RUO were smaller than the other three angles. The means of ∠RBI and ∠RLO were larger than the other three angles. All the angles measured by the two observers were significantly different from 0. There was no significant difference between the means of ∠RUO and ∠RIT, while there were significant differences between ∠RIT and the other two angles.

### 3.7. The Measurement Result for Non-Elderly Subjects

Table 9 and Table 10 are the results for the non-elderly subjects. Non-elderly subjects were defined as subjects younger than 65 years of age. The means and standard deviations of ∠RIT and ∠RUO were smaller than the other three angles. The means of ∠RBI were the largest among the five angles. All the angles measured by the two observers were significantly different from 0. The means of ∠RUO and ∠RIT showed no significant differences in all three measurements, ∠RSC and ∠RIT showed no significant differences in one measurement, while ∠RBI and ∠RLO both showed significant differences compared with ∠RIT.

### 3.8. The Measurement Result for Elderly Subjects

Table 11 and Table 12 are the results for the elderly subjects. The means and standard deviations of ∠RIT and ∠RUO were smaller than the other three angles. The means of ∠RBI were the largest among the five angles. All the angles measured by the two observers were significantly different from 0. The means of ∠RUO and ∠RIT showed no significant differences in two measurements, ∠RSC and ∠RIT showed no significant differences in one measurement, while ∠RBI and ∠RLO both showed significant differences compared with ∠RIT.

### 3.9. Outliers Analysis

Figure 5 is the boxplot of the five different angles in three measurements. The data of ∠RBI were the most dispersed, followed by ∠RLO and ∠RSC, while the data of ∠RIT and ∠RUO were relatively more concentrated. The outliers of ∠RBI were all greater than 3 degrees. The outliers of ∠RLO and ∠RSC were all greater than 2 degrees. The outliers of ∠RIT and ∠RUO were almost all greater than 1.5 degrees. Table 13 is the result of our analysis on the overall data after removing the two data points with the largest deviation of the angle for the 5 TAPs. The means and standard deviations of ∠RIT and ∠RUO were smaller than the other three angles. The means of ∠RBI were the largest among the five angles. After removing outliers, all five angles still remained significantly different from 0. The means of ∠RUO and ∠RIT showed no significant differences in all three measurements, while there are significant differences between ∠RIT and the other three angles. Table 14 compares the measurement results of ∠RIT with four other angles, showing that only the measurement results of ∠RUO and ∠RIT have no significant difference, while the other three angles are significantly different from ∠RIT. In Table 15, we analyzed the outliers we removed and compiled their imaging characteristics.

## 4. Discussion

We found that ∠RIT and ∠RUO had smaller means and standard deviations in the overall sample, different gender groups, and different age groups, indicating that the IT line and UOF were the two TAPs with higher precision. Moreover, the mean values of ∠RIT and ∠RUO did not show significant differences in results except for the elderly subgroup. ∠RBI had the largest means in the overall sample, different gender groups, and different age groups, indicating that BI line is the least precise among the 5 TAPs. The mean values of ∠RSC and ∠RLO fell between ∠RUO and ∠RBI. None of the mean values of the angles differed significantly from 0, indicating that the five TAPs are not perfectly parallel to the IFH line. In the outlier analysis, we found that the data distribution of RIT and RUO was the most concentrated. However, after removing some outliers, the mean values of each angle were still significantly different from 0. The ICC values of ∠RIT and ∠RUO were lower than that of other angles. Despite the lower reliability of these two angles, both still have a moderate level of reliability.

The teardrop is recognized as the best landmark for measuring acetabular cup position [24,25,26]. Tucker et al. pointed out that the teardrop has good interobserver reliability in the range of 7.5° of rotation, 30° inlet tilt, and 15° of outlet tilt [24]. Bayraktar et al. indicated that the teardrop is a more consistent pelvis landmark than ischial tuberosity when measuring the acetabular inclination angle [9]. Schofer et al. summarized six methods for locating the hip rotation center and found that the method using teardrops as the landmark could most closely locate the true anatomical center of the femoral head [30]. Massin et al. pointed out that the teardrop can best measure the horizontal and vertical migration of the acetabular cup [26]. The migration in these two directions is related to the femoral offset and LLD, respectively. In our study, we also found that the measurement results of the IT line are relatively more accurate compared with the other TAPs.

Despite its accuracy, the teardrop is often not clear, making it incompetent as a landmark. According to our data, 34.2% of pelvic AP radiographs cannot use the teardrop as a pelvic landmark. In addition, our clinical experience shows that the teardrop may be worn away during the THA operation, making it less effective as a landmark after surgery. In this regard, our data showed that radiographs with acetabular components had a slightly higher proportion of teardrops that are not clear compared with radiographs without acetabular components (36% versus 33.9%). It is important to find other alternative landmarks since more than 1/3 of patients who underwent THA cannot use teardrops as a landmark for postoperative evaluation.

In our research, we found that the UOF is also a good pelvic landmark, probably because of its closeness to the teardrop. By definition, the lower rim of the teardrop is adjacent to the UOF [25]. Massin et al. indicated that when the teardrop is not visible, the obturator line (UOF) could be used to measure the vertical migration of the acetabular component [26]. Boudriot et al. found that the intersection between Koehler’s line (ilioischial line) and a line between the upper rims of the two obturator foramens (UOF) is a reliable landmark to determine the anatomic hip center [32]. Wegner et al. used a line drawn through the top of the obturator foramen (UOF) to replace the IT line when the base of the teardrop was not clearly visible [33]. Our multiple measurements by different observers also demonstrate that the UOF exhibits accuracy no less than the IT line in both overall data and measurement results in the subdivided subgroups. Furthermore, the probability of being unidentifiable is much lower for the UOF compared with the IT line (1% versus 34.2%). However, it is worth mentioning that since part of the obturator foramen might be blocked by the nearby bony structure, the measurer should carefully locate the UOF.

The ICC of the measurements of ∠RIT and ∠RUO were lower compared with the other three angles in the assessment of intra- and interobserver reliability. We speculate that this was because these two angles were smaller, making the ICC of these two angles more sensitive to the measurement differences. Another possible reason for their lower reliability is that the lower edge of the teardrops and the upper edge of the obturator foramen are less distinct compared with the other three pelvic landmarks. Despite being less reliable than the other three angles, measurements of ∠RIT and ∠RUO still exhibit a moderate level of reliability.

Identifying the causes of these outliers can help us clarify the conditions for applying each TAP. In Table 15 of the outlier analysis section, we reviewed the pelvic AP radiographs of the outliers in an attempt to identify the cause(s) of these outliers. One of the outliers of the SC line is caused by the uneven protrusion of the iliac crest. The two outliers of the IT line are related to the asymmetrical shape of the teardrops on both sides of the pelvis. One of the outliers of the RUO is related to the deformation of the obturator foramen in the image. We suspect that the occurrence of other outliers may be due to measurement errors or the anatomical characteristics of the subject’s pelvis.

As for the difference in the five included angles between the five TAPs and the reference axis, we suspect it is caused by image distortion, pelvic asymmetry, and the uneven protrusions on the pelvis landmark. Pelvic AP radiographs are susceptible to 3D variations of pelvic orientation such as pelvic tilt, obliquity, and rotation, resulting in image distortion [34,35]. Pelvic asymmetry refers to the iliac crests on both sides being unequal in distance from the acetabulum [36]. Pelvic asymmetry is not just pathological; it can occur in healthy people without functional abnormalities [37]. We speculate that teardrops are less susceptible to image distortion and pelvic asymmetry due to their smaller size, while larger structures such as iliac crests and ischial tuberosities tend to be affected by these two factors. In addition, there are often some uneven small protrusions on iliac crests and ischial tuberosities, making some outliers occasionally appear on the measurements of the ∠RSC and the ∠RBI.

There are significant differences in pelvic anatomy between males and females. These differences also exist in several anatomic landmarks related to TAP. Harris et al. found significant differences in the distance between the lateral border of the teardrop and the anteromedial insertion of the transverse acetabular ligament between the sexes [38]. Bombaci et al. measured the vertical and horizontal distances from the hip rotation center to the inter-ischial line (BI line) and teardrop, respectively, and found that although there were significant differences in the quantitative values of the aforementioned distances between the sexes, the ratios of these distances relative to pelvic height had no significant gender differences [39]. Despite gender differences in pelvic anatomy, our study shows that the IT line and UOF exhibit greater accuracy in measurement results than the other three TAP lines.

In our study, there are slight differences in different age groups. Not many studies focused on the relationship between age and pelvic anatomy. In the study done by Boudriot et al., when using teardrops as the reference point to find the hip rotation center, the results of the group over 60 years old were significantly different from the other age groups [32]. Although age may affect the pelvic structure and the measurements of the pelvic axis, our study shows that the IT line and UOF exhibit greater accuracy in measurement results than the other three TAP lines.

In addition to the postoperative evaluation, we believe that the outcome of our study also has the potential to assist in the development of more appropriate physiotherapy. Physiotherapy after THA is crucial for the restoration of the patient’s functional outcome. It can help build hip abductor strength and improve mobility, gait speed, and cadence after THA [40,41]. The TAP can help physicians assess the severity of LLD and pelvic inclination of the patients after THA, which can affect the patient’s motor function and muscle strength [23]. Perhaps future research can try to link postoperative evaluation of the THA with the choices of different physical therapy regimens and intensities in order to formulate more precise physiotherapy plans.

The clinical significance of this study lies in establishing the appropriate TAPs and their applicable conditions. Orthopedic surgeons were previously unsure which pelvic landmark to use as a reference point for drawing TAPs when the teardrops were unclear. Orthopedic surgeons may occasionally use TAPs other than the IT line for measurements, but there is currently no comprehensive research that elucidates the accuracy of these alternative TAPs. Our study clarified the deviation between each TAP and the IFH line and also identified the possible reasons for outliers in each axis measurement. Orthopedic surgeons can perform postoperative evaluations of LLD, hip rotation center, and pelvic inclination angle measurements in THA based on our recommendations regarding which TAP to use in which situation (see Appendix A).

There were some limitations in our study. The patients we selected were not completely healthy patients, but patients without severe hip abnormalities on pelvic AP radiographs. Although the orthopedic diseases in these patients had not yet caused significant damage to the anatomy of the hip joint, they may still have some impact on the measurements. Additionally, our research on which axis is more representative of the TAP was based on the pelvis AP radiograph of patients without the acetabular component. However, we believe that the same axis will still be useful in patients after THA. Acetabular implants are installed in the hip joints, while the landmarks that make up the TAPs are located on the pelvis. Therefore, the presence of acetabular implants should not have a significant impact on the effectiveness of the TAPs.

## 5. Conclusions

In conclusion, we recommend using the IT line as the TAP when the lower boundary of the teardrops is clear and the shapes of the teardrops on both sides of the pelvis are symmetrical. When there is no deformation of the obturator foramen on pelvic AP radiographs, the UOF is also a good choice for the TAP. If the aforementioned two axes are difficult to delineate, then the SC line and UOF will be the next consideration for the TAP. We do not recommend the BI line as the TAP.

## Figures and Tables

**Figure 1 biomedicines-11-01397-f001:**
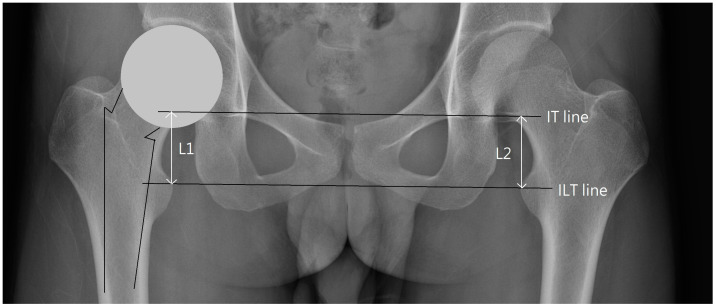
The current method used to measure LLD involves comparing the difference between the distances of the inter-teardrops line (IT line) and the inter-lesser trochanters line (ILT line) on both sides (i.e., the difference between L1 and L2). When the teardrops are not clearly visible, it is currently uncertain which TAP to use for measuring LLD.

**Figure 2 biomedicines-11-01397-f002:**
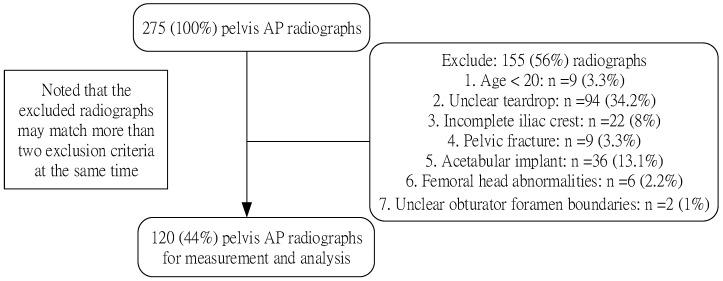
Flow diagram of the recruitment process.

**Figure 3 biomedicines-11-01397-f003:**
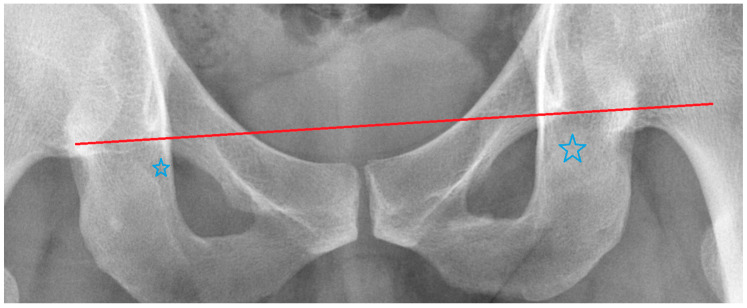
Location of the UOF. Carefully trace along the edge of the obturator foramen, the UOF should be the red line. This happened because part of the obturator foramen (marked with blue stars) was obscured by the bone in front.

**Figure 4 biomedicines-11-01397-f004:**
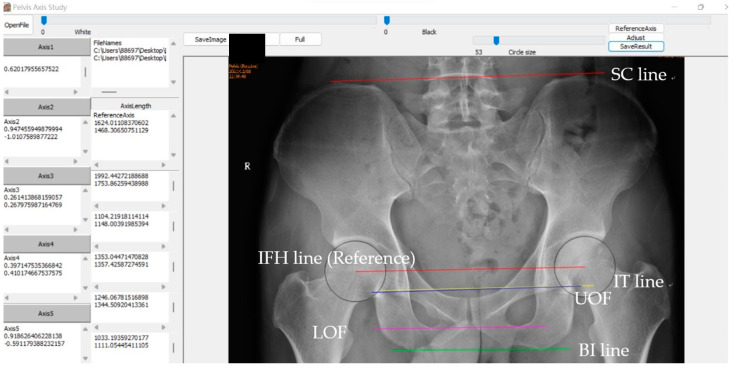
The axes drawn using the software “PelvicAxis”. Red line: SC line. Green line: BI line. Yellow line: IT line. Blue line: UOF. Purple line: LOF. Red line: IFH line (Reference Axis).

**Figure 5 biomedicines-11-01397-f005:**
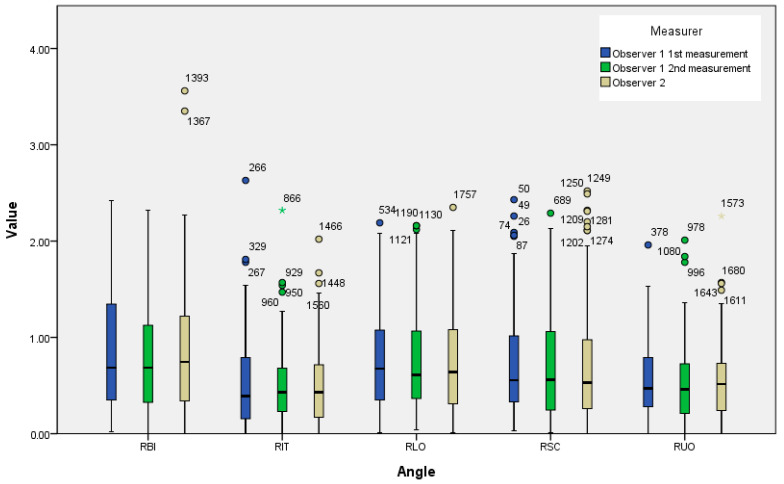
Boxplots of the five different angles in three measurements. The numbers of outliers in the figure were the numbers of the data listed in the statistical software, which has no special meaning.

**Table 1 biomedicines-11-01397-t001:** Basic information of the subjects.

Group	Value
Total	120 (100%)
Gender	
Male	58 (48.3%)
Female	62 (51.7%)
Age	
34≤	26 (21.7%)
35–44	15 (12.5%)
45–54	18 (15%)
55–64	15 (12.5%)
65–74	23 (19.2%)
≥75	23 (19.2%)
Mean	56.1
Standard deviation	21.2

**Table 2 biomedicines-11-01397-t002:** The ICC in the measurements of each axis.

	ICC ^1^	Reliability ^2^
**Intraobserver**		
∠RSC	0.945	Excellent
∠RBI	0.905	Excellent
∠RIT	0.749	Moderate
∠RUO	0.725	Moderate
∠RLO	0.920	Excellent
**Interobserver**		
∠RSC	0.920	Excellent
∠RBI	0.850	Good
∠RIT	0.679	Moderate
∠RUO	0.617	Moderate
∠RLO	0.913	Excellent

^1^ ICC: intra- and interobserver correlation coefficient. ICC value greater than 0.9 suggests excellent reliability; between 0.75 and 0.9 suggests a good reliability; ^2^ between 0.5 and 0.75 suggests a moderate reliability; below 0.5 suggests a poor reliability.

**Table 3 biomedicines-11-01397-t003:** The measurement result for all subjects.

Angle	First Measurement of Observer 1	Second Measurement of Observer 1	Observer 2
∠RSC	0.709° ± 0.554°	0.697° ± 0.576°	0.707° ± 0.612°
	*p* = 0.000	*p* = 0.000	*p* = 0.000
∠RBI	0.876° ± 0.630°	0.796° ± 0.582°	0.844° ± 0.667°
	*p* = 0.000	*p* = 0.000	*p* = 0.000
∠RIT	0.517° ± 0.454°	0.481° ± 0.374°	0.498° ± 0.396°
	*p* = 0.000	*p* = 0.000	*p* = 0.000
∠RUO	0.562° ± 0.401°	0.507° ± 0.384°	0.542° ± 0.382°
	*p* = 0.000	*p* = 0.000	*p* = 0.000
∠RLO	0.736° ± 0.496°	0.738° ± 0.498°	0.756° ± 0.548°
	*p* = 0.000	*p* = 0.000	*p* = 0.000

**Table 4 biomedicines-11-01397-t004:** The comparison of the mean of ∠RIT with the other four angles (all subjects).

Hypothesis	First Measurement of Observer 1	Second Measurement of Observer 1	Observer 2
H_0_: µ_∠RSC =_ µ_∠RIT_ H_1_: µ_∠RSC_ ≠ µ_∠RIT_	*p =* 0.002	*p =* 0.001	*p =* 0.003
H_0_: µ_∠RBI_ = µ_∠RIT_ H_1_: µ_∠RBI_ ≠ µ_∠RIT_	*p =* 0.000	*p =* 0.000	*p =* 0.000
H_0_: µ_∠RUO =_ µ_∠RIT_ H_1_: µ_∠RUO_ ≠ µ_∠RIT_	*p =* 0.370	*p =* 0.555	*p =* 0.307
H_0_: µ_∠RLO =_ µ_∠RIT_ H_1_: µ_∠RLO_ ≠ µ_∠RIT_	*p =* 0.000	*p =* 0.000	*p =* 0.000

**Table 5 biomedicines-11-01397-t005:** The measurement results for female subjects.

Angle	First Measurement of Observer 1	Second Measurement of Observer 1	Observer 2
∠RSC	0.668° ± 0.527°	0.686° ± 0.558°	0.662° ± 0.593°
	*p* = 0.000	*p* = 0.000	*p* = 0.000
∠RBI	0.874° ± 0.563°	0.822° ± 0.594°	0.873° ± 0.638°
	*p* = 0.000	*p* = 0.000	*p* = 0.000
∠RIT	0.504° ± 0.445°	0.465° ± 0.339°	0.516° ± 0.404°
	*p* = 0.000	*p* = 0.000	*p* = 0.000
∠RUO	0.617° ± 0.434°	0.544° ± 0.406°	0.536° ± 0.330°
	*p* = 0.000	*p* = 0.000	*p* = 0.000
∠RLO	0.675° ± 0.482°	0.692° ± 0.515°	0.716° ± 0.544°
	*p* = 0.000	*p* = 0.000	*p* = 0.000

**Table 6 biomedicines-11-01397-t006:** The comparison of the mean of ∠RIT with the other four angles (female subjects).

Hypothesis	First Measurement of Observer 1	Second Measurement of Observer 1	Observer 2
H_0_: µ_∠RSC =_ µ_∠RIT_ H_1_: µ_∠RSC_ ≠ µ_∠RIT_	*p* = 0.056	*p =* 0.006	*p =* 0.126
H_0_: µ_∠RBI_ = µ_∠RIT_ H_1_: µ_∠RBI_ ≠ µ_∠RIT_	*p =* 0.000	*p =* 0.000	*p =* 0.000
H_0_: µ_∠RUO =_ µ_∠RIT_ H_1_: µ_∠RUO_ ≠ µ_∠RIT_	*p =* 0.095	*p =* 0.197	*p =* 0.721
H_0_: µ_∠RLO =_ µ_∠RIT_ H_1_: µ_∠RLO_ ≠ µ_∠RIT_	*p =* 0.024	*p =* 0.002	*p =* 0.011

**Table 7 biomedicines-11-01397-t007:** The measurement result for male subjects.

Angle	First Measurement of Observer 1	Second Measurement of Observer 1	Observer 2
∠RSC	0.754° ± 0.583°	0.708° ± 0.599°	0.755° ± 0.633°
	*p* = 0.000	*p* = 0.000	*p* = 0.000
∠RBI	0.878° ± 0.699°	0.767° ± 0.573°	0.813° ± 0.701°
	*p* = 0.000	*p* = 0.000	*p* = 0.000
∠RIT	0.531° ± 0.467°	0.498° ± 0.410°	0.479° ± 0.391°
	*p* = 0.000	*p* = 0.000	*p* = 0.000
∠RUO	0.503° ± 0.357°	0.468° ± 0.359°	0.549° ± 0.434°
	*p* = 0.000	*p* = 0.000	*p* = 0.000
∠RLO	0.802° ± 0.507°	0.786° ± 0.479°	0.798° ± 0.553°
	*p* = 0.000	*p* = 0.000	*p* = 0.000

**Table 8 biomedicines-11-01397-t008:** The comparison of the mean of ∠RIT with the other four angles (male subjects).

Hypothesis	First Measurement of Observer 1	Second Measurement of Observer 1	Observer 2
H_0_: µ_∠RSC =_ µ_∠RIT_ H_1_: µ_∠RSC_ ≠ µ_∠RIT_	*p* = 0.011	*p =* 0.036	*p =* 0.009
H_0_: µ_∠RBI_ = µ_∠RIT_ H_1_: µ_∠RBI_ ≠ µ_∠RIT_	*p =* 0.002	*p =* 0.003	*p =* 0.003
H_0_: µ_∠RUO =_ µ_∠RIT_ H_1_: µ_∠RUO_ ≠ µ_∠RIT_	*p =* 0.718	*p =* 0.639	*p =* 0.296
H_0_: µ_∠RLO =_ µ_∠RIT_ H_1_: µ_∠RLO_ ≠ µ_∠RIT_	*p =* 0.005	*p =* 0.001	*p =* 0.001

**Table 9 biomedicines-11-01397-t009:** The measurement results for non-elderly subjects.

Angle	First Measurement of Observer 1	Second Measurement of Observer 1	Observer 2
∠RSC	0.733° ± 0.604°	0.697° ± 0.580°	0.714° ± 0.639°
	*p* = 0.000	*p* = 0.000	*p* = 0.000
∠RBI	0.838° ± 0.618°	0.783° ± 0.595°	0.858° ± 0.611°
	*p* = 0.000	*p* = 0.000	*p* = 0.000
∠RIT	0.496° ± 0.481°	0.474° ± 0.399°	0.528° ± 0.437°
	*p* = 0.000	*p* = 0.000	*p* = 0.000
∠RUO	0.539° ± 0.396°	0.528° ± 0.412°	0.517° ± 0.351°
	*p* = 0.000	*p* = 0.000	*p* = 0.000
∠RLO	0.718° ± 0.457°	0.729° ± 0.473°	0.781° ± 0.526°
	*p* = 0.000	*p* = 0.000	*p* = 0.000

**Table 10 biomedicines-11-01397-t010:** The comparison of the mean of ∠RIT with the other four angles (non-elderly subjects).

Hypothesis	First Measurement of Observer 1	Second Measurement of Observer 1	Observer 2
H_0_: µ_∠RSC =_ µ_∠RIT_ H_1_: µ_∠RSC_ ≠ µ_∠RIT_	*p =* 0.006	*p =* 0.011	*p =* 0.050
H_0_: µ_∠RBI_ = µ_∠RIT_ H_1_: µ_∠RBI_ ≠ µ_∠RIT_	*p =* 0.000	*p =* 0.000	*p =* 0.001
H_0_: µ_∠RUO =_ µ_∠RIT_ H_1_: µ_∠RUO_ ≠ µ_∠RIT_	*p =* 0.509	*p =* 0.355	*p =* 0.839
H_0_: µ_∠RLO =_ µ_∠RIT_ H_1_: µ_∠RLO_ ≠ µ_∠RIT_	*p =* 0.004	*p =* 0.001	*p =* 0.002

**Table 11 biomedicines-11-01397-t011:** The measurement results for elderly subjects.

Angle	First Measurement of Observer 1	Second Measurement of Observer 1	Observer 2
∠RSC	0.670° ± 0.468°	0.696° ± 0.577°	0.696° ± 0.573°
	*p* = 0.000	*p* = 0.000	*p* = 0.000
∠RBI	0.938° ± 0.649°	0.814° ± 0.567°	0.822° ± 0.754°
	*p* = 0.000	*p* = 0.000	*p* = 0.000
∠RIT	0.550° ± 0.410°	0.493° ± 0.334°	0.448° ± 0.318°
	*p* = 0.000	*p* = 0.000	*p* = 0.000
∠RUO	0.599° ± 0.411°	0.473° ± 0.336°	0.583° ± 0.428°
	*p* = 0.000	*p* = 0.000	*p* = 0.000
∠RLO	0.765° ± 0.558°	0.751° ± 0.541°	0.715° ± 0.585°
	*p* = 0.000	*p* = 0.000	*p* = 0.000

**Table 12 biomedicines-11-01397-t012:** The comparison of the mean of ∠RIT with the other four angles (elderly subjects).

Hypothesis	First Measurement of Observer 1	Second Measurement of Observer 1	Observer 2
H_0_: µ_∠RSC =_ µ_∠RIT_ H_1_: µ_∠RSC_ ≠ µ_∠RIT_	*p* = 0.118	*p =* 0.021	*p =* 0.021
H_0_: µ_∠RBI_ = µ_∠RIT_ H_1_: µ_∠RBI_ ≠ µ_∠RIT_	*p =* 0.001	*p =* 0.001	*p =* 0.001
H_0_: µ_∠RUO =_ µ_∠RIT_ H_1_: µ_∠RUO_ ≠ µ_∠RIT_	*p =* 0.546	*p =* 0.763	*p =* 0.042
H_0_: µ_∠RLO =_ µ_∠RIT_ H_1_: µ_∠RLO_ ≠ µ_∠RIT_	*p =* 0.031	*p =* 0.003	*p =* 0.006

**Table 13 biomedicines-11-01397-t013:** The measurement result for the overall data after removing the outliers.

Angle	First Measurement of Observer 1	Second Measurement of Observer 1	Observer 2
∠RSC	0.669° ± 0.497°	0.655° ± 0.529°	0.654° ± 0.549°
	*p* = 0.000	*p* = 0.000	*p* = 0.000
∠RBI	0.826° ± 0.605°	0.734° ± 0.528°	0.771° ± 0.550°
	*p* = 0.000	*p* = 0.000	*p* = 0.000
∠RIT	0.482° ± 0.394°	0.460° ± 0.324°	0.479° ± 0.378°
	*p* = 0.000	*p* = 0.000	*p* = 0.000
∠RUO	0.550° ± 0.386°	0.501° ± 0.370°	0.529° ± 0.349°
	*p* = 0.000	*p* = 0.000	*p* = 0.000
∠RLO	0.695° ± 0.464°	0.698° ± 0.463°	0.715° ± 0.507°
	*p* = 0.000	*p* = 0.000	*p* = 0.000

**Table 14 biomedicines-11-01397-t014:** The comparison of the mean of ∠RIT with the other four angles (after removing the outliers).

Hypothesis	First Measurement of Observer 1	Second Measurement of Observer 1	Observer 2
H_0_: µ_∠RSC =_ µ_∠RIT_ H_1_: µ_∠RSC_ ≠ µ_∠RIT_	*p =* 0.002	*p* = 0.002	*p =* 0.014
H_0_: µ_∠RBI_ = µ_∠RIT_ H_1_: µ_∠RBI_ ≠ µ_∠RIT_	*p =* 0.000	*p =* 0.000	*p =* 0.000
H_0_: µ_∠RUO =_ µ_∠RIT_ H_1_: µ_∠RUO_ ≠ µ_∠RIT_	*p =* 0.156	*p =* 0.297	*p =* 0.221
H_0_: µ_∠RLO =_ µ_∠RIT_ H_1_: µ_∠RLO_ ≠ µ_∠RIT_	*p =* 0.000	*p =* 0.000	*p =* 0.000

**Table 15 biomedicines-11-01397-t015:** The outliers we removed and their image features.

The Outliers We Removed	Image Features
Outlier 1 of ∠RSC	Uneven protrusions on bilateral iliac crests.
Outlier 2 of ∠RSC	Normal
Outlier 1 of ∠RBI	Poor image quality, but borders of the pelvic landmarks are still visible.
Outlier 2 of ∠RBI	Right side femur fracture without femoral head dislocation.
Outlier 1 of ∠RIT	Asymmetrical teardrop shapes on both sides.
Outlier 2 of ∠RIT	Asymmetrical teardrop shapes on both sides.
Outlier 1 of ∠RUO	Normal
Outlier 2 of ∠RUO	The obturator foramen is slightly distorted on the image.
Outlier 1 of ∠RLO	Normal
Outlier 2 of ∠RLO	Normal

## Data Availability

The data sets used and/or analyzed during the present study are available from the corresponding author upon request.

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
