# Peer review of "The Transverse Mechanical Axis of the Pelvis for Post-Operative Evaluation of Total Hip Arthroplasty"

_biomedicines, 2023, doi:10.3390/biomedicines11051397_

Round 1
Reviewer 1 Report
Thank you for the opportunity to read this interesting paper. The authors investigated the best TAP among the five candidates of the TAPs on normal hip radiographs. The authors concluded that IT line and UOF line are both appropriate. However, there are some concerns including methodology to be revised.
<Major comments>
・Evaluation items: absolute value/SD/outlier
I think absolute value of the difference is more important than mean value. Positive or negative values of difference is meaningless in this study described at line 119. The authors should evaluate and analyze absolute values.
As for research for accuracy like this study, the data of outlier are also important. The authors should define the outlier and analyze it.
There are many sentences that SDs are smaller/larger. However, was the comparison between the SDs analyzed statistically?
・Clinical significance
Please discuss a clinical significance of this study based on the differences observed in this study.
I think that measurement difference of the inclination angle or LLD such as 1 to 2 degrees, millimeters does not influence clinical practice.
I understand the difference among the five angles. However, dose the difference change clinical practice? Or there is a difference but is the difference clinically trivial?
<Minor comments>
・ICCs
ICCs should be shown not last but first in the Results section, because the ICCs are quite important fundamentals in reading the results.
・Figure 1 and Table 2
The reasons of exclusion are seven in Figure 1 and eight in Table 2. This discrepancy should be corrected.
Figure 1 and Table 2 present the same information. Either one should be shown.
Reviewer 2 Report
- This manuscript offers little original and useful content to the orthopaedic clinician or radiologist in the assessment of implant position.
- References can be a bit more up to date. Most older than 2018.
- Study design: If the teardrop line is used for the post-operative assessment of THA then surely the measurements are made on an AP radiograph including the acetabular implant. So why were the radiographs that show an acetabular implant excluded?
- Are we certain that when the teardrop is not visibly obvious that the other axes can be confidently drawn or are they also affected? Maybe include those cases as they were excluded, to show that this method is also valid in those non-ideal cases.
- Don’t understand lines 105-108.
- Reference axis is IFH line — is that validated?
- Line 140-141?
- Results needs improved illustration. Needs graphical representation of the results (box plot with statistical significance shown).
- Bland-Altman?
- Wrong choice of words in some cases.
- ‘Moderate’ reliability for the recommended axes… Is this good enough as the reliability is lower?
- Line 292-293?
Round 2
Reviewer 1 Report
Thank you for response to my comments and revising the manuscript.
The authors have responded correctly and have revised appropriately.
I think your manuscript is suitable to be published in Biomedicines.
I appreciate your efforts to re-analyze your data and improve your manuscript.
Reviewer 2 Report
I acknowledge the changed made by the authors, however, I'm not convinced that this body of work adds anything of value to the literature.
Another coordinate system on plain radiographs is not particularly helpful.